# MAPLE: Context-aware Multimodal Augmentation for Long-tail 3D Object Detection

## Abstract

3D object detection is essential for autonomous driving but remains limited by the long-tail distribution of real-world data. Instance-level augmentation methods, whether copy-paste or asset rendering, are typically restricted to LiDAR and offer only modest variation with limited scene context. We introduce **MAPLE**, a training-free pipeline for multimodal augmentation that generates synchronized RGB–LiDAR pairs. Objects are inserted through context-aware inpainting in the image domain, and paired pseudo-LiDAR is reconstructed via depth estimation. To ensure cross-modal plausibility, MAPLE incorporates semantic and geometric verification modules that filter inconsistent generations. We further propose a success-rate evaluation that quantifies error reduction across verification stages, providing a principled measure of pipeline reliability. On the nuScenes benchmark, MAPLE consistently improves both detection and segmentation in multimodal and LiDAR-only settings. We will release code to support reproducibility

## 1 Introduction

Reliable 3D perception (Chen et al., 2023; Yin et al., 2021; Lang et al., 2019; Yan et al., 2018; Zhou & Tuzel, 2018; Wang et al., 2023) is critical for autonomous driving (Caesar et al., 2020; Geiger et al., 2012) and robotics (Cadena et al., 2016; Zhang et al., 2014), yet progress is limited by the long-tail distribution of real-world data. As shown in Table 1, rare but safety-critical categories such as construction vehicles, bicycles, and motorcycles account for less than one percent of large-scale benchmarks, leading to degraded recognition (Zhu et al., 2019; Yaman et al., 2023). Instance-level augmentation has been proposed to mitigate this imbalance without additional data (Yan et al., 2018). While these methods provide measurable gains, copy-paste or curated-asset rendering still yield limited variation (Yan et al., 2018; Chang et al., 2024). The emergence of foundation models offers a promising alternative, as vision-language models (VLMs) and diffusion models can generate diverse content. Text3DAug (Reichardt et al., 2024) follows this direction, using lightweight textual descriptions (e.g., red sports car) to guide 3D asset generation (Jun & Nichol, 2023). Despite improved diversity, these pipelines remain LiDAR-only and offer limited scene context in placement.

We introduce **MAPLE**, a training-free foundation pipeline that generates synchronized RGB-LiDAR pairs. Unlike unimodal approaches, MAPLE inserts objects through context-guided image inpainting, without relying on external map annotations such as semantic labels. VLMs provide diverse object descriptions (Fig. 3), and diffusion models generate them into varied visual and

Table 1: Imbalanced classes in nuScenes.

| Class | Ratio (%) | Class | Ratio (%) |
|---|---|---|---|
| Car | 42.30 | Trailer | 2.13 |
| Pedestrian | 17.86 | Bus | 1.40 |
| Barrier | 13.04 | **Constr. veh.** | 1.26 |
| Traf. cone | 8.40 | **Motorcycle** | 1.08 |
| Truck | 7.59 | **Bicycle** | 1.02 |

geometric forms (Fig. 4), together expanding intra-class variation for rare categories. These image-level generations are then paired with LiDAR through a depth estimator, producing geometrically consistent point clouds. Achieving plausibility and structural fidelity across modalities is challenging; MAPLE addresses this with semantic and geometric verification, which filter misaligned or artifact-heavy generations and preserve scale consistency. As illustrated in Fig. 1, our generated samples (red) align with surrounding landmarks (green), whereas the existing LiDAR-only method (Chang et al., 2024) uses road-semantics-based placement. Finally, we propose a success-rate evaluation to measure the reliability of generative augmentation. By showing how semantic and geometric verification reduce implausible generations, our protocol quantifies improvements in the effective yield of usable samples under the same generation budget. Beyond MAPLE, this reliability analysis

Figure 1: We present **MAPLE**, a multimodal instance augmentation pipeline for safety-critical long-tail classes. Objects are first inserted into the image via context-aware inpainting and then projected into LiDAR as geometrically plausible counterparts. In the middle example, the generated object (red box 1) is integrated between two original objects (green boxes 1-2) in the image, and its paired LiDAR instance appears at a consistent location. In contrast, existing LiDAR-only instance augmentation uses road-semantics-based placement, which limits scene awareness. Green boxes (1-4) indicate original objects (landmarks), while red boxes (1-2) indicate MAPLE-generated samples.

provides a principled tool for evaluating generative augmentation pipelines, supporting the reliable use of synthesized data.

Our contributions are threefold:

- We present MAPLE, the first training-free framework for multimodal instance augmentation to address long-tail imbalance, and introduce a success-rate evaluation for assessing the reliability of its synthesized data.

- We design verification modules that promote semantic alignment and geometric plausibility in the synthesized RGB-LiDAR pairs.

- We demonstrate on the nuScenes benchmark that MAPLE consistently improves 3D object detection and semantic segmentation in both multimodal and LiDAR-only settings.

## 2 RELATED WORK

**Instance-Level Augmentation for Long-tail Problem.** Existing driving datasets (Geiger et al., 2012; Caesar et al., 2020; Sun et al., 2020) remain limited in scale, leading to generalization difficulties and long-tail imbalance. Scene-level augmentation approaches address this by oversampling scenes that contain rare classes (Zhu et al., 2019; Yaman et al., 2023; Gupta et al., 2019), whereas instance-level augmentation directly increases the frequency of rare-class objects (Yan et al., 2018; Zhan et al., 2023; Šebek et al., 2022; Chang et al., 2024). A central challenge in instance-level augmentation lies in placement: determining where and how to insert objects so that they appear natural within the scene. Prior work has adopted different strategies, including random insertion, learned placeability maps, and semantic-label-based rules. However, these pipelines remain unimodal, operating only on LiDAR and relying on semantic-label-driven placement, which makes limited use of scene context. In contrast, our multimodal augmentation pipeline inserts objects with visual context-aware guidance, enabling realistic placement without requiring external annotation maps.

**Automated Augmentation Pipelines for Perception Tasks.** Recent advances in foundation models–diffusion models, vision-language models (VLMs), and large language models (LLMs)–have enabled more diverse and higher-quality data generation (Rombach et al., 2022; OpenAI, 2023). Leveraging these models, recent work has proposed training-free augmentation pipelines that compose foundation models into automated systems for increasing data diversity (Kupyn & Rupprecht, 2024). In 2D perception, combining diffusion with VLMs has shown that such modular pipelines can enrich training distributions and improve downstream performance (Wu et al., 2023; Wang et al., 2024a). Similar ideas have been explored in 3D perception, where generative models synthesize novel LiDAR objects or replace head classes with rare classes (Reichardt et al., 2024; Yurt et al., 2025). However, these efforts remain unimodal and thus unexplore the importance of cross-modal consistency. Building on this trajectory, we introduce MAPLE, a training-free pipeline for multimodal augmentation that generates paired RGB-LiDAR samples through context-aware inpainting and depth-based pairing, while ensuring reliability via semantic and geometric verification. Beyond this, we present a success-rate evaluation that provides a general framework for assessing the robustness of generative augmentation.

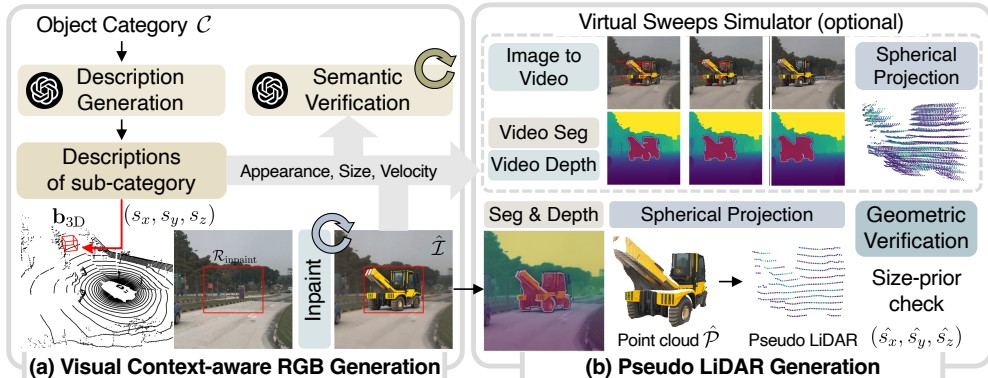

**(a) Visual Context-aware RGB Generation**  **(b) Pseudo LiDAR Generation**

Figure 2: **Overview of MAPLE.** (a) Visual context-aware RGB generation. An LLM suggests subclass descriptions with appearance, size, and motion cues, while projected 3D boxes define inpainting regions. Diffusion then synthesizes objects directly into the scene, followed by semantic verification to discard invalid results. (b) Pseudo-LiDAR generation. The verified images are converted into depth and projected to form scaled pseudo-LiDAR objects. A virtual sweep simulator can further expand them into multi-sweep sequences. Geometric verification filters out implausible structures, yielding paired RGB-LiDAR augmentations that are both visually coherent and physically consistent.

## 3 METHOD

Our goal is to design a training-free generative augmentation pipeline that increases both the frequency and diversity of rare classes, while maintaining visual-context awareness and geometric plausibility between RGB images and LiDAR point clouds. Beyond generating new objects, it is essential to evaluate the pipeline's effective yield–that is, the proportion of synthesized samples that remain usable after verification. To address this, we introduce a success-rate evaluation protocol (Section 4) that quantifies yield improvements and demonstrates how semantic and geometric verification progressively reduce errors and preserve reliable samples throughout the pipeline.

Fig. 2 provides an overview of our framework, MAPLE. In Section 3.1, we describe rare-class object generation in the RGB domain through inpainting guided by vision-language and diffusion models, followed by semantic verification to ensure contextual plausibility. In Section 3.2, we explain how paired LiDAR samples are reconstructed from depth estimation, augmented with a virtual sweep simulator, and filtered through geometric verification to enhance structural plausibility.

### 3.1 RGB GENERATION WITH SEMANTIC VERIFICATION

The first stage of MAPLE operates in the image domain, synthesizing rare-class objects as the basis for multimodal augmentation. Its objectives are twofold: to expand intra-class diversity and to ensure context-aware placement so that inserted objects blend naturally with the surrounding scene. This is achieved by the VLM proposing candidate appearances and approximate sizes, subsequently realized through diffusion-based generation. Finally, semantic verification filters out samples that lack semantic consistency or contextual plausibility. Implementation details are provided in Section E.2.

**LLM-based object description.** We query a VLM OpenAI (2023) to provide a plausible subclass of the target label, a short visual description, its typical physical size, and an estimated velocity range.

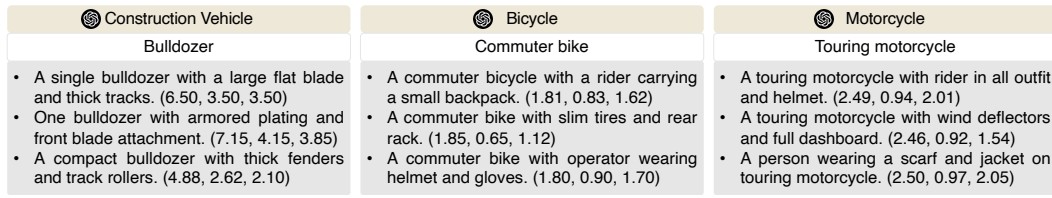

Figure 3: **Examples of LLM-generated object descriptions.** Even within a single target category, the LLM outputs diverse descriptions varying in subclass identity and appearance. The first row shows the target class, the second row representative subclasses, and each entry specifies traits with estimated average size $(s_x, s_y, s_z)$, which guide inpainting and provide priors for geometric checks.

Figure 4: **Diffusion-based inpainting with visual context-aware placement.** (a) Inpainted objects are placed with plausible heading and scale, blending with nearby structures and exhibiting varied shapes, appearances, and realistic occlusions. (b) By contrast, naive 2D box placement often yields implausible results, such as construction vehicles larger than traffic lights or buildings.

```
  Provide one subclass of {TARGET_LABEL} commonly found in urban environments. Include
a short visual description (e.g., shape, notable features), its typical physical size
(length, width, and height), and the expected velocity range in m/s based on real-world
usage. For the size, identify a specific real-world product model from the web and use
its official specifications. If the target label is "bicycle" or "motorcycle," generate
two separate descriptions: (1) without a rider, reporting the vehicle's dimensions
only, and (2) with a rider seated, reporting the bounding box dimensions that include
both the vehicle and the person.
```

As illustrated in Fig. 3, the resulting descriptions $\mathcal{T}_c$ expand intra-class variation beyond the dataset and guide subsequent stages: they condition the inpainting process, serve as references for semantic and geometric verification, and provide motion priors for virtual sweep simulation.

**Diffusion-based inpainting with 3D-to-2D projection.** We first sample candidate 3D bounding boxes $\mathbf{b}_{3D} = (c_x, c_y, c_z, s_x, s_y, s_z, \theta)$, where poses $(c_x, c_y, c_z, \theta)$ are drawn uniformly and sizes $(s_x, s_y, s_z)$ follow priors from $\mathcal{T}_c$. Each box is projected into the image plane using camera intrinsics $\mathbf{K}$ and extrinsics $[\mathbf{R} \mid \mathbf{t}]$, yielding a 2D mask $\mathcal{M}$. A candidate is retained only if the projected region does not overlap excessively with existing objects, enforced by requiring its intersection-over-union (IoU) with any annotated instance to remain below a threshold. To maintain visual context awareness, we extract an image patch centered on the inpainting region $\mathcal{R}_{\text{inpaint}}$ while preserving the surrounding context. When the resolution of this patch is insufficient, we upsample it with a super-resolution model $f_{\text{SR}}$ (Yue et al., 2024). A diffusion-based inpainting model $f_{\text{in}}$ (Zhuang et al., 2024; Ju et al., 2024), conditioned on the textual description $\mathcal{T}c$ and mask $\mathcal{M}$, synthesizes the object as $\hat{\mathcal{I}} = f_{\text{in}}(f_{\text{SR}}(\mathcal{I}_{\text{patch}}), \mathcal{T}_c, \mathcal{M})$. When the inpainted object fully occupies $\mathcal{R}_{\text{inpaint}}$, its dimensions align with the intended 3D scale specified by $\mathcal{T}_c$. Although $\theta$ is sampled uniformly, the inpainting process is influenced by the surrounding context when determining the orientation and details.

**Semantic verification.** Each inpainted image $\hat{\mathcal{I}}$ is evaluated through semantic verification to ensure that only valid generations proceed to the multimodal pipeline. The process uses the conditioning description $\mathcal{T}_c$ and the inpainting region $\mathcal{R}_{\text{inpaint}}$ coordinates, and applies two checks:

```
  Given the inpainted image {IMG_TOKEN}, the intended description {PROMPT}, and the
inpainting region coordinates {REGION}, answer the following:
(1) Does the object correspond to the intended subclass?
(2) Within the region, is there exactly one instance, and does it fully occupy it?
Answer format: [Yes, Yes] / [No, Yes] / [Yes, No] / [No, No]
```

Because inpainting attempts are not always successful, we generate multiple candidates and retain only those that satisfy both conditions. Our iterative sampling-and-filtering strategy follows the spirit of RANSAC (Derpanis, 2010), where inconsistent hypotheses are progressively discarded until stable inliers remain. As a result, mislabeled, undersized, or cluttered augmentations are removed before pseudo-LiDAR reconstruction, and the retained samples provide semantically reliable instances for downstream alignment. Examples of the resulting context-aware generations are shown in Fig. 4.

### 3.2 PSEUDO-LiDAR GENERATION WITH GEOMETRIC VERIFICATION

We then construct paired LiDAR samples from depth estimation. Given the projected box and the estimated object height, we apply depth scaling to align the pseudo-LiDAR with the intended

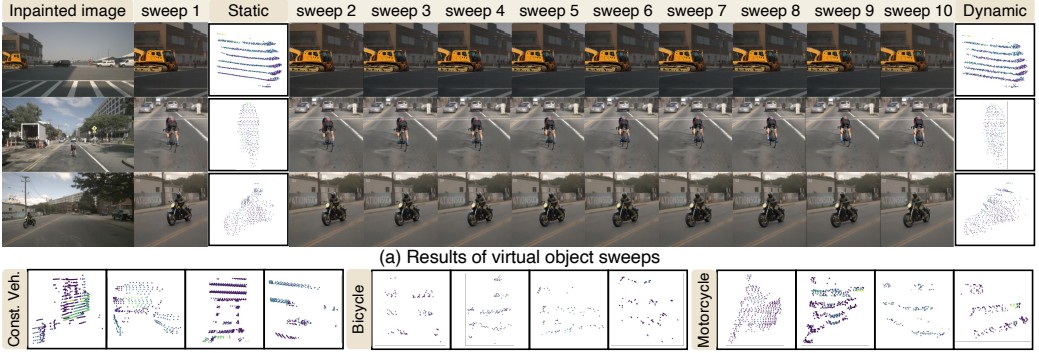

Figure 5: **RGB-LiDAR instance composition.** We show augmented RGB images with their 1-sweep pseudo-LiDAR reconstructions, which closely resemble real sensor measurements. For comparison, nuScenes instances confirm that scale-normalized reconstruction yields pseudo-LiDAR objects consistent with both images and ground-truth point clouds.

(a) Results of virtual object sweeps

(b) Data distribution across 10 LiDAR sweeps

Figure 6: **Results of virtual object sweeps with 10-sweep pseudo-LiDAR.** In the static setting, only ego-motion is applied, while the dynamic setting simulates both ego- and object motion to yield realistic trajectories. The bottom row shows real 10-sweep nuScenes point clouds, confirming that our generated data preserves geometric consistency with real LiDAR.

physical size provided by the VLM description. To further support modern benchmarks, which rely on multi-sweep LiDAR accumulation to mitigate sparsity, we optionally introduce a virtual sweep simulator that extends the reconstructed pseudo-LiDAR across multiple temporal sweeps with visual-context awareness. After this step, geometric verification ensures structural plausibility by rejecting objects with inconsistent scales or structures. This process constrains that augmented objects not only appear realistic in the RGB domain but also remain physically consistent in the LiDAR domain. Implementation details are provided in the Section E.3.

**Depth-based reconstruction.** Given an inpainted image $\hat{\mathcal{I}}$, our goal is to recover a 3D object representation consistent with the intended physical size. We begin by applying a semantic segmentator (Ren et al., 2024) to isolate the generated object region $\mathcal{R}_{\text{obj}} \subset \hat{\mathcal{I}}$. A monocular depth estimator (Wang et al., 2024b) then predicts a dense depth map, which is unprojected with the camera intrinsics $\mathbf{K}$ to form a raw point cloud. Finally, the cloud is rescaled to match the prior height $s_z$ specified by the VLM description: $\hat{\mathcal{P}} = \{\frac{s_z}{\hat{s}_z} \cdot \hat{D}(u,v) \mathbf{K}^{-1}[u,v,1]^T \mid (u,v) \in \mathcal{R}_{\text{obj}}\}$, where $\hat{s}_z$ is the predicted object height before scaling. While the prior box specifies $(s_x, s_y, s_z, \theta)$, in practice, only the vertical extent $s_z$ serves as a reliable anchor for metric scaling, as the 3D-to-2D projection enforces that the inpainted object fully occupies the designated region along the vertical axis. The remaining dimensions and orientation are left flexible so that the inpainting process can adapt width, length, and heading to the surrounding scene, preserving geometric consistency in height while maintaining visual plausibility in the horizontal layout. To match real LiDAR sampling, $\hat{\mathcal{P}}$ is transformed into spherical coordinates $(r, \theta, \phi)$, discretized to the angular resolution of the target sensor, and clipped to its field of view. The range image $\hat{\mathcal{P}}_{\text{lidar}}$ is then unprojected back to 3D, yielding a pseudo-LiDAR object that matches the beam and scanline density of the real sensor.

**Virtual sweep simulation.** To align with multi-sweep benchmarks (Caesar et al., 2020; Li et al., 2023a), we extend each pseudo-LiDAR object $\hat{\mathcal{P}}_{\text{lidar}}$ into a temporally consistent sequence. nstead of replicating point clouds with random headings and velocities (Chang et al., 2024; Yan et al., 2018),

we generate motion seeds that respect both the inpainted image $\hat{\mathcal{I}}$ and the scaled pseudo-LiDAR $\tilde{\mathcal{P}}$. Class-specific velocity priors from $\mathcal{T}_c$ define coarse displacements $\delta_t$ applied to the point cloud, i.e., $\hat{\mathcal{P}}_t = \hat{\mathcal{P}} + t \cdot \mathbf{v}$, where $\mathbf{u}_t^{(k)} = \pi(\mathbf{x}_t^{(k)})$ for each point $\mathbf{x}_t^{(k)} \in \hat{\mathcal{P}}_t$. The image-to-video diffusion model (Niu et al., 2024) conditions

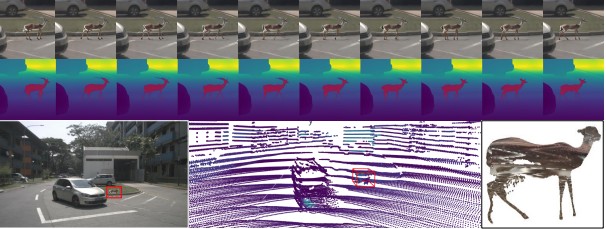

Figure 7: **Multi-sweep generation.** Our method generates temporally coherent pseudo-LiDAR across multiple sweeps.

on $\hat{\mathcal{I}}$ and $\{\mathbf{u}_t^{(k)}\}$ to synthesize temporally coherent RGB sequences, which are then converted into per-sweep pseudo-LiDAR $\hat{\mathcal{P}}_{\text{lidar}}^t$ (Chen et al., 2025; Ravi et al., 2024). As illustrated in Fig. 6 and Fig. 7, this step extends single-frame generations into multi-sweep trajectories, maintaining consistency with benchmark protocols and adding realism such as animal legs or bicycle pedaling.

**Geometric verification.** Although depth-based reconstruction provides scale normalization, monocular estimators introduce boundary noise. To improve plausibility, we apply two filters. First, *spatial filtering* retains the densest region of points around the centroid in the $XY$ plane. The retained fraction $D$ is determined empirically: experiments reveal that larger $D$ improves fidelity for smaller objects, whereas smaller $D$ is adequate for larger ones, reflecting the tendency of depth estimators to produce less precise boundaries for thin objects. Second, a *size-prior check* enforces $\lambda_{\min} s_i \leq \hat{s}_i \leq \lambda_{\max} s_i, i = x, y, z$, ensuring consistency between the reconstructed dimensions $\hat{s}_i$ and the priors $s_i$ from $\mathcal{T}_c$. Together, these filters suppress depth artifacts and unrealistic reconstructions, ensuring that only reliable pseudo-LiDAR objects are used for downstream training.

## 4 SUCCESS-RATE EVALUATION OF GENERATIVE PIPELINES

We introduce a reliability evaluation to assess the robustness of our generative pipeline. This evaluation provides a metric for the production efficiency of the pipeline, establishing MAPLE as the first training-free framework for reliable multimodal augmentation in long-tail 3D perception.

We first define the naive failure rate without verification as

$$\mathbb{P}(F_{\text{naive}}) = \mathbb{P}(F_{\text{inpaint}}) + \mathbb{P}(F_{\text{depth}})(1 - \mathbb{P}(F_{\text{inpaint}})), \tag{1}$$

where $\mathbb{P}(F_{\text{inpaint}})$ and $\mathbb{P}(F_{\text{depth}})$ denote failure rates of the inpainting stage and depth estimation stage.

After adding verification modules, the verified failure rate becomes

$$\mathbb{P}(F_{\text{verified}}) = \mathbb{P}(F_{\text{inpaint}}^{\text{after filtered}}) + \mathbb{P}(F_{\text{depth}}^{\text{after filtered}})\big(1 - \mathbb{P}(F_{\text{inpaint}}^{\text{after filtered}})\big) \tag{2}$$

$$= \mathbb{P}(F_{\text{inpaint}})(1 - \alpha_{\text{sem}}) + \mathbb{P}(F_{\text{depth}})(1 - \alpha_{\text{geo}})\big[1 - \mathbb{P}(F_{\text{inpaint}})(1 - \alpha_{\text{sem}})\big], \tag{3}$$

where $\alpha_{\text{sem}}$ and $\alpha_{\text{geo}}$ denote the removal rates of semantic and geometric verification, respectively. Here, $\mathbb{P}(F_{\text{depth}}^{\text{after filtered}})$ denotes the failure rate of the size-prior check after spatial filtering, ensuring that only structurally consistent reconstructions are tested against category-specific priors.

As shown in Table 2, this two-stage verification protocol substantially reduces total failure rates, ensuring that only reliable samples are passed to downstream perception. In practice, $\alpha_{\text{sem}}$ is estimated from 1,000 human-annotated images, while $\alpha_{\text{geo}}$ is computed as $1 - \big[\mathbb{P}(F_{\text{depth}}^{\text{after filtered}})/\mathbb{P}(F_{\text{depth}})\big]$.

Table 2: **Sequential failure analysis.** For each category, we report the number of rejected objects out of all synthesized objects (# Rej. / Total). The overall failure rates $\mathbb{P}(F_{\text{naive}})$ and $\mathbb{P}(F_{\text{verified}})$ highlight how semantic and geometric verification substantially reduce error propagation. All values are reported with 95% confidence intervals.

| Category | # of Rej. / Total | $\mathbb{P}(F_{\text{inpaint}})$ | $\mathbb{P}(F_{\text{depth}})$ | $\mathbb{P}(F_{\text{depth}}^{\text{after filtered}})$ | $\mathbb{P}(F_{\text{naive}})$ | $\mathbb{P}(F_{\text{verified}})$ |
|---|---|---|---|---|---|---|
| Constr. Veh. | 62,941 / 262,584 | $19.34\% \pm 0.12$ | $41.23\% \pm 0.27$ | $0.83\% \pm 0.02$ | 52.6% | **10.4%** |
| Bicycle | 24,399 / 490,453 | $4.74\% \pm 0.06$ | $37.05\% \pm 0.21$ | $1.66\% \pm 0.03$ | 39.7% | **2.4%** |
| Motorcycle | 13,235 / 583,553 | $2.22\% \pm 0.04$ | $34.15\% \pm 0.20$ | $0.90\% \pm 0.02$ | 35.6% | **1.1%** |

# 5 EXPERIMENT

## 5.1 IMPLEMENTATION DETAILS

Our pipeline leverages pretrained 2D foundation models. Module-wise costs are reported in Table 3, with model details in Section E. We augment three rare categories–*construction vehicle*, *motorcycle*, and *bicycle*–on the nuScenes benchmark (Caesar et al., 2020), inserting up to seven construction vehicles and five motorcycles or bicycles per scene, while common classes follow standard GT-Aug (Yan et al., 2018). To ensure plausibility, candidate regions $\mathcal{R}_{\text{inpaint}}$ are discarded if overlapping with previously inserted instances beyond a fixed IoU threshold. For LiDAR-only settings, augmented data are merged with the nuScenes ground-truth database, comparable to Text3DAug (Reichardt et al., 2024) and PGT-Aug (Chang et al., 2024). Training configurations are given in Section F.

Table 3: **Module-wise computational cost.** Parameter counts (M) and runtime per sample measured on A100 GPUs.

| Module | Params (M) | Time (s) |
|---|---|---|
| Inpainter | 2075.4 | 5.08 |
| GPT(description) | – | 0.01 |
| GPT(verification) | – | 0.65 |
| ImageSAM | 224.5 | 0.14 |
| Grounding Model | 172.8 | – |
| Image Depth | 314.2 | 0.59 |
| **Total (1-sweep)** | **2811.9** | **6.47** |
| Image-to-Video | 2948.8 | 11.13 |
| VideoSAM | 224.5 | 1.90 |
| Video Depth | 384.4 | 2.09 |
| **Total (10-sweep)** | **6378.3** | **21.59** |

## 5.2 MULTIMODAL 3D PERCEPTION

We first assess the downstream impact of MAPLE in multimodal settings. For 3D object detection we adopt BEVFusion (Liang et al., 2022), and for 3D semantic segmentation we use 2DPASS (Yan et al., 2022). To our knowledge, MAPLE is among the first augmentation frameworks that synthesize paired RGB-LiDAR samples for rare categories, allowing multimodal training data to be expanded in a cross-consistent manner. As shown in Table 4, MAPLE improves performance on rare categories while maintaining accuracy on frequent ones. The effect is particularly pronounced for construction vehicles, where both detection and segmentation benefit from the increased intra-class diversity. We also investigate more challenging scenarios, including zero-shot detection and extended rare-class evaluation (Section B and Section C). These settings are rarely covered in prior work due to the small number of evaluation samples. Although the absolute performance remains modest, MAPLE provides initial evidence that multimodal augmentation can generalize beyond standard benchmarks.

## 5.3 UNIMODAL 3D PERCEPTION

Since the final output of MAPLE is pseudo-LiDAR, we also evaluate its effectiveness in LiDAR-only settings. For detection we use CenterPoint (Yin et al., 2021) and PointPillar (Lang et al., 2019), and for segmentation we use MinkowskiNet (Choy et al., 2019) and SPVCNN (Tang et al., 2020). As summarized in Table 5, MAPLE performs on par with GT-Aug, Text3DAug, and PGT-Aug for detection, and achieves the strongest mIoU across segmentation backbones. Unlike these LiDAR-specific baselines, MAPLE originates from a multimodal pipeline yet remains competitive in the unimodal domain. We attribute the segmentation gains to context-aware placement and verification, which improve boundary labeling where networks rely heavily on surrounding cues (Vora et al., 2020; Qiu et al., 2025). These results confirm that MAPLE's pseudo-LiDAR preserves geometric plausibility and supports both detection and segmentation.

## 5.4 MULTIMODAL QUALITY EVALUATION

To compare with prior LiDAR-only augmentation methods, we evaluate pseudo-LiDAR quality under the 1-sweep setting. Qualitative examples are shown in Section D. We adopt two metrics in the embedding space of an SE(3)-Transformer (Fuchs et al., 2020) trained on nuScenes: (i) FID between real and synthesized instance embeddings (Chang et al., 2024) and (ii) feature entropy, measuring intra-class diversity. We report results per category, restricting to instances with at least 64 points.

Table 4: **Multimodal 3D perception results.** We report mAP for detection and mIoU for segmentation, including per-class results for three rare categories MAPLE enhances rare-class performance in both detection and segmentation without degrading overall accuracy.

| Detection | mAP | Constr. Veh. | Bicycle | MC | Segmentation | mIoU | Constr. Veh. | Bicycle | MC |
|---|---|---|---|---|---|---|---|---|---|
| BEVFusion | 64.27 | 27.99 | 56.63 | 71.32 | 2DPASS | 77.65 | 56.83 | 47.78 | 85.72 |
| + Ours | **65.37** | **29.31** | **58.88** | **71.82** | + Ours | **78.15** | **59.77** | **48.74** | **86.44** |

Table 5: **LiDAR-only 3D perception results.** Evaluated with LiDAR-only detectors and segmenters, MAPLE demonstrates effective transfer of pseudo-LiDAR augmentation, yielding consistent segmentation improvements and competitive detection accuracy relative to LiDAR-only baselines.

| Detection | mAP | Constr. Veh. | Bicycle | MC | Segmentation | mIoU | Constr. Veh. | Bicycle | MC |
|---|---|---|---|---|---|---|---|---|---|
| CenterPoint | | | | | MinkowskiNet | | | | |
| + GT-Aug | 62.57 | 22.62 | 57.52 | 68.25 | + GT-Aug | 72.01 | 25.29 | 28.45 | 73.33 |
| + Text3DAug | _62.81_ | **24.23** | 57.65 | **69.86** | + Text3DAug | 73.27 | 43.00 | 24.93 | 81.98 |
| + PGT-Aug | 62.68 | 23.41 | _59.07_ | 68.69 | + PGT-Aug | 73.50 | 38.03 | 36.90 | 80.47 |
| + Ours | **62.95** | _23.99_ | **59.20** | 68.77 | + Ours | **74.31** | **45.48** | **39.05** | **82.58** |
| PointPillar | | | | | SPVCNN | | | | |
| + GT-Aug | _50.93_ | 19.11 | 19.34 | **51.98** | + GT-Aug | 73.85 | 37.64 | 40.69 | 75.59 |
| + Text3DAug | 50.85 | 20.12 | _21.95_ | 47.67 | + Text3DAug | 73.80 | 41.87 | 34.74 | 81.01 |
| + PGT-Aug | **50.96** | **21.94** | 20.04 | 49.85 | + PGT-Aug | 73.80 | 42.01 | 37.56 | 82.08 |
| + Ours | 50.83 | _20.41_ | **22.12** | _51.41_ | + Ours | **75.02** | **50.02** | **41.51** | **83.17** |

Table 7: **Ablation on semantic and geometric verification.** FID of pseudo-LiDAR for three rare classes under different module settings. Applying both checks yields the lowest FID, with clear gains for slender objects such as bicycles and motorcycles.

| | Constr. Veh. | | | | Bicycle | | | | Motorcycle | | | |
|---|---|---|---|---|---|---|---|---|---|---|---|---|
| Semantic Verification | – | ✓ | – | ✓ | – | ✓ | – | ✓ | – | ✓ | – | ✓ |
| Geometric Verification | – | – | ✓ | ✓ | – | – | ✓ | ✓ | – | – | ✓ | ✓ |
| FID | 7.44 | 7.44 | **6.61** | **6.61** | 2.93 | 2.77 | 0.91 | **0.86** | 9.14 | 9.10 | 5.15 | **5.10** |

**Fidelity.** As shown in Table 6 and Fig. 9, MAPLE achieves lower FID scores than mesh-based baselines (PGT-Aug, Text3DAug), reflecting greater similarity to the structural patterns of real objects. This trend holds across both large categories such as construction vehicles and smaller ones, with bicycles showing the most notable improvements and motorcycles performing slightly below PGT-Aug but still stronger than Text3DAug.

**Diversity.** UMAP embeddings (Fig. 8a) show that MAPLE samples overlap with real instances, reflecting both diversity and semantic coherence. Our method achieves higher entropy than PGT-Aug across categories, indicating that our RGB generation framework enriches intra-class diversity by producing broader object appearances than manually collected datasets with limited examples. For construction vehicles and motorcycles, however, entropy is slightly lower than Text3DAug. As illustrated in (Fig. 8b), Text3DAug and PGT-Aug insert objects without scene context, resulting in grid-like patterns in the bird's-eye view. By contrast, MAPLE places objects in a visually context-aware manner, often closer to the ego-vehicle where they are visible in the image domain–a distribution consistent with LiDAR's distance-dependent characteristics. Despite these differences, our synthesized data achieves competitive downstream performance in 3D perception tasks. This suggests that even with slightly lower entropy, MAPLE's context-aware placement offers clear benefits for scene-centric applications, such as semantic segmentation.

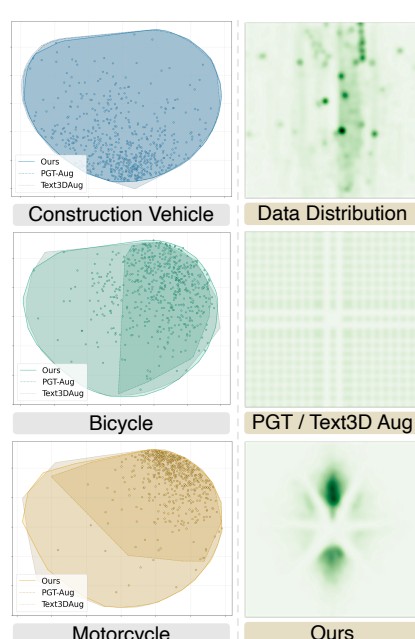

Figure 8: **(a)** UMAP of LiDAR instance features. Convex hulls show synthesized coverage; dots denote real data. **(b)** BEV distributions within $[-25.6, 25.6]$ m. Densities normalized: GT by its own distribution; others by the maximum across methods.

Table 8: **Effect of semantic verification on 3D object detection.** Training with semantic verification(Sem. Verif.) improves both multimodal (BEVFusion) and unimodal (CenterPoint) performance.

| Multimodal | mAP | Constr. Veh. | Bicycle | MC | Unimodal | mIoU | Constr. Veh. | Bicycle | MC |
|---|---|---|---|---|---|---|---|---|---|
| Ours | **65.37** | **29.31** | **58.88** | **71.82** | Ours | **62.95** | **23.99** | **59.20** | **68.77** |
| - Sem. Verif. | 64.81 | 27.89 | 56.07 | 70.98 | - Sem. Verif. | 62.94 | 23.50 | 57.58 | 68.07 |

Table 6: **1-sweep RGB–LiDAR quality.** LiDAR FID and UMAP feature entropy per category.

| Class | Method | FID (↓) | Entropy (↑) |
|---|---|---|---|
| Constr. Veh. | Text3DAug | 6.84 | **6.12** |
| | PGT-Aug | 7.60 | 5.93 |
| | Ours | **6.61** | 6.00 |
| Bicycle | Text3DAug | 12.07 | 5.66 |
| | PGT-Aug | 2.10 | 5.72 |
| | Ours | **0.86** | **5.98** |
| MC | Text3DAug | 9.21 | **6.04** |
| | PGT-Aug | **3.70** | 5.59 |
| | Ours | 5.10 | 5.96 |

Figure 9: **Qualitative 1-sweep pseudo-LiDAR.** MAPLE is comparable to mesh-based baselines.

Figure 10: **Rejected examples** during semantic verification.

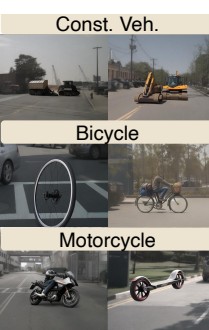

## 5.5 THE EFFECT OF VERIFICATION MODULES

Verification ensures that generated samples are both diverse and reliable for downstream perception, as shown by ablation studies highlighting the complementary roles of semantic and geometric checks.

**Semantic verification.** This step addresses category mismatch, incomplete or undersized inpaintings that fail to cover the designated region, and multi-instance artifacts that can be produced by diffusion models (Fig. 10). As shown in Table 8, semantic verification improves detection in both multimodal and LiDAR-only settings, with more pronounced gains for multimodal detectors sensitive to visual noise. It thus serves as a quality control stage that removes inconsistent hypotheses, preserves diverse yet valid samples, and improves the pipeline's effective yield, as shown by our success-rate evaluation.

**Geometric verification.** After semantic filtering, depth-based reconstruction can still cause scale distortion, boundary noise, or implausible shapes. This is critical for slender categories such as bicycles and motorcycles, where small depth errors lead to large distortions. By applying spatial filtering and size priors, geometric verification removes unrealistic structures and keeps objects within plausible bounds. As seen in Table 7, this step improves fidelity for thin objects and further boosts usable–sample rates in success-rate evaluation.

**Discussion.** The two modules are complementary: semantic verification ensures category alignment and enables context-aware insertion, while geometric verification corrects depth artifacts and provides a bridge to the LiDAR domain by maintaining structural consistency. Our sequential failure analysis confirms these roles, showing that errors are progressively reduced and usable sample rates increase across stages. Limitations remain, such as VLM hallucinations (Li et al., 2023b) and the sim-to-real gap in LiDAR intensity (Viswanath et al., 2024; Marcus et al., 2025). Nevertheless, our results indicate that the pipeline offers a practical step toward training-free generative augmentation for multimodal long-tail perception. Looking ahead, MAPLE's modular design allows components to be replaced as foundation models advance, while success-rate evaluation offers a way to measure how such improvements translate into higher yields of reliable samples.

## 6 CONCLUSION

We presented **MAPLE**, a training-free, verification-aware framework that synthesizes paired RGB–LiDAR instances for long-tail 3D perception. Unlike unimodal augmentation methods, MAPLE enables visual context-aware insertion, producing objects that blend naturally with their surroundings while remaining geometrically consistent across modalities. The central idea is to couple such context-aware generation with semantic and geometric verification, ensuring that augmentation not only increases the occurrence of rare classes but also enhances their diversity and reliability. We further propose a success-rate evaluation that quantifies effective yield and tracks error reduction across stages, enabling systematic measurement of independent modules within the pipeline. Its modular design also allows components to be updated as foundation models advance, with success-rate evaluation offering a principled means to assess how such upgrades translate into improved yield. Finally, experiments on nuScenes confirm that MAPLE improves detection and segmentation in both multimodal and LiDAR-only settings.

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

## A   APPENDIX

In this appendix, we provide extended experiments, qualitative results, and implementation details of **MAPLE**. Section B reports a challenging zero-shot experiment on the *bicycle* class, designed to stress-test the framework's ability to recover recognition ability for categories entirely absent from training. Section C presents extended results on rare but safety-critical categories—such as *animal* and *police vehicle*—that are not evaluated under the standard 10-class nuScenes protocol. Section D provides additional qualitative examples illustrating how MAPLE synthesizes diverse and contextually consistent instances. Finally, Section E and Section F include implementation details of the pipeline and training configurations for the 3D perception tasks.

## B   ZERO-SHOPT EXPERIMENTAL RESULTS

Table 9: **Zero-shot detection results for the bicycle category (0.5 m threshold).** All bicycle instances were removed from the training set. GT-Aug, despite reusing 8,185 real bicycles 1.69M times, completely fails. In contrast, MAPLE is trained without access to a single real bicycle instance, yet achieves the first successful detections, demonstrating its ability to recover previously unseen categories purely through generative augmentation.

| Method | AP | P@R=0.1 | P@R=0.2 | P@R=0.3 | P@R=0.5 | P@R=0.7 |
|--------|-------|---------|---------|---------|---------|---------|
| GT-Aug | 0.000 | 0.000 | 0.000 | 0.000 | 0.000 | 0.000 |
| Ours | **8.021** | **0.399** | **0.118** | 0.000 | 0.000 | 0.000 |

We design an adversarial *zero-shot* setting to stress-test MAPLE's ability to recover categories that are entirely absent from training. Specifically, all bicycle instances—the rarest class in nuScenes—were removed from the training set. The GT-Aug baseline, despite reusing 8,185 real bicycle objects and inserting them 1.69M times across 22,461 scenes, collapses to 0 mAP. In contrast, MAPLE, without access to a single real bicycle instance, generates 514,852 synthetic bicycles and achieves the first non-trivial recognition ability on this category (8.0 mAP, 0.4 precision at 10% recall).

While the absolute AP is understandably modest under such an adversarial condition, the key result is that MAPLE transforms a complete failure case into a detectable signal. Even a small but consistent set of true positives is practically valuable: it provides a bootstrap for downstream active learning and annotation pipelines, and demonstrates that generative augmentation can go beyond database sampling by introducing novel, diverse instances. This highlights MAPLE's unique potential to inject recognition capability in extreme scarcity regimes where conventional augmentation entirely fails.

## C   EXTENDED EXPERIMENTAL RESULTS

Table 10: **Class distribution of nuScenes annotations.** Each column group shows the number of cuboids and their ratio for a given class.

| Category | # Cuboids | Ratio (%) | Category | # Cuboids | Ratio (%) | Category | # Cuboids | Ratio (%) |
|----------|-----------|-----------|----------|-----------|-----------|----------|-----------|-----------|
| **animal** | 787 | 0.07 | personal mobility | 395 | 0.03 | barrier | 152,087 | 13.04 |
| adult | 208,240 | 17.86 | police officer | 727 | 0.06 | debris | 3,016 | 0.26 |
| child | 2,066 | 0.18 | stroller | 1,072 | 0.09 | pushable | 24,605 | 2.11 |
| construction worker | 9,161 | 0.79 | wheelchair | 503 | 0.04 | traffic cone | 97,959 | 8.40 |
| bike rack | 2,713 | 0.23 | **bicycle** | 11,859 | 1.02 | bus.bendy | 1,820 | 0.16 |
| bus.rigid | 14,501 | 1.24 | car | 493,322 | 42.30 | **construction** | 14,671 | 1.26 |
| ambulance | 49 | 0.00 | **police** | 638 | 0.05 | **motorcycle** | 12,617 | 1.08 |
| trailer | 24,860 | 2.13 | truck | 88,519 | 7.59 | | | |

The nuScenes dataset (Caesar et al., 2020) defines 23 object categories for 3D bounding box annotations, covering a broad spectrum of urban traffic objects. However, as shown in Table 10, the distribution of annotated cuboids is highly imbalanced. To mitigate this long-tail skew, the official benchmark consolidates 23 categories into 10 major classes for 3D detection. While this improves label balance, it also prevents explicit evaluation of rare but safety-critical categories. To address

Table 12: **Per-class detection performance on extended rare categories under multimodal training (0.5 m threshold).** MAPLE substantially improves recognition across all rare categories, including *animal* and *police vehicle*, which are excluded from the official nuScenes protocol. Whereas baselines often fail to detect such ultra-rare classes, MAPLE introduces meaningful recognition ability by injecting diverse synthetic instances, directly increasing true positives (TPs) and demonstrating scalability beyond the standard benchmark taxonomy.

| Category | Method | mAP | Prec@r=0.5 | Prec@r=0.7 |
|---|---|---|---|---|
| Motorcycle | BEVFusion | 0.819 | 0.953 | 0.864 |
|  | + Ours | **0.871** | **0.973** | **0.942** |
| Bicycle | BEVFusion | 0.723 | 0.915 | 0.587 |
|  | + Ours | **0.826** | **0.964** | **0.897** |
| Constr. Veh. | BEVFusion | 0.671 | 0.435 | 0.000 |
|  | + Ours | **0.815** | **0.815** | 0.000 |
| Police | BEVFusion | 0.809 | 0.928 | 0.828 |
|  | + Ours | **0.937** | **0.987** | **0.987** |
| Animal | BEVFusion | 0.416 | 0.296 | 0.135 |
|  | + Ours | **0.612** | **0.666** | **0.411** |

this gap, we extend the evaluation set to include two underrepresented classes—*animal* and *police vehicle*—that are excluded from the standard protocol.

Table 11: **Multimodal 3D object detection performance on the extended train/val splits.** We report class-wise AP with a minimum precision and recall threshold of 0. Our method augments **objects** to address class imbalance and supports arbitrary categories without manual annotations.

| | Car | Truck | Constr. Veh. | Bus | Trailer | Barrier | Motorcycle | Bicycle | Pedestrian | Traffic Cone | Animal | Police | mAP |
|---|---|---|---|---|---|---|---|---|---|---|---|---|---|
| *Val Set* | | | | | | | | | | | | | |
| BEVFusion | 0.890 | 0.602 | **0.380** | 0.724 | 0.450 | **0.759** | 0.739 | 0.560 | 0.883 | 0.806 | 0.0003 | 0.016 | 0.567 |
| + Ours | **0.897** | **0.627** | 0.371 | **0.752** | **0.461** | 0.754 | **0.751** | **0.635** | **0.895** | **0.814** | **0.0014** | **0.071** | **0.586** |
| *Train Set* | | | | | | | | | | | | | |
| BEVFusion | 0.912 | 0.703 | 0.671 | 0.832 | 0.670 | 0.884 | 0.819 | 0.723 | 0.920 | 0.893 | 0.416 | 0.809 | 0.771 |
| + Ours | **0.932** | **0.821** | **0.815** | **0.904** | **0.777** | **0.925** | **0.871** | **0.826** | **0.949** | **0.924** | **0.612** | **0.937** | **0.868** |

We further evaluate MAPLE on an extended nuScenes benchmark that explicitly includes rare but safety-critical categories—*animal* and *police vehicle*—which are excluded from the official protocol. As shown in Table 10, these categories account for only 0.07% and 0.05% of annotations, respectively, making standard validation metrics unstable. To mitigate this, we additionally report class-wise results on the training set, where such categories appear more frequently. While training-set performance may partly reflect memorization, it still reveals whether models can recognize novel categories introduced by augmentation.

As summarized in Table 12, MAPLE consistently improves recognition across all long-tail categories, including construction vehicles, bicycles, motorcycles, animals, and police vehicles. Notably, even for ultra-rare categories, MAPLE substantially increases both precision and mAP, with gains reflected directly in the number of true positives (TPs) recovered. Whereas baselines often yield zero detections under extreme scarcity, MAPLE introduces meaningful recognition ability by injecting diverse synthetic instances with varied shapes, scales, and poses. This demonstrates that our framework scales beyond the official taxonomy and promotes more robust decision boundaries even in severely low-annotation regimes.

## D  ADDITIONAL QUALITATIVE RESULTS

This section presents additional qualitative results generated by MAPLE across underrepresented object categories. MAPLE easily supports new categories because it can flexibly extend to arbitrary target classes without requiring additional manual annotations. Here, we showcase examples involving animals (Fig. 11) and police vehicles (Fig. 12), extending construction vehicles (Fig. 13), motorcycles (Fig. 14), and bicycles (Fig. 15). Our visualizations highlight the effectiveness of context-aware instance placement, which ensures that synthesized objects are seamlessly integrated into the scene with realistic scale and spatial alignment.

We show: (a) the inpainted RGB image, where the inserted object is synthesized at a context-aware scale and position, ensuring visual and geometric consistency with the surrounding scene. (b) a sequence of video frames synthesized for 10-sweep simulation, providing temporally coherent motion. (c) per-frame depth estimation and semantic segmentation results, used to construct pseudo-LiDAR point clouds and verify object identity and boundaries. (d) the resulting 1-sweep pseudo-LiDAR, showing the point cloud of the inserted instance from a single frame. (e) the accumulated 10-sweep pseudo-LiDAR, capturing temporally propagated objects via our motion simulation module.

Specifically, in (e), we visualize pseudo-LiDARs under static and dynamic settings. The static setting assumes the object remains fixed at its $t = 0$ location, with only ego-motion applied. The dynamic setting simulates both ego and object motion, resulting in a realistic object trajectory across frames. Note that for *motorcycle* and *bicycle* instances without riders, our motion simulation module is not applied. In these cases, we generate multi-sweep pseudo-LiDARs only under the static setting.

## E  IMPLEMENTATION DETAILS

### E.1  PRETRAINED FOUNDATION MODELS USED IN OUR PIPELINE

Our pipeline leverages a set of pretrained 2D foundation models, each specialized for a different sub-task in the multimodal data generation process.

**Models for One-Sweep Generation.**    We utilize several foundation models to generate high-quality multimodal data in a single-frame setting:

- **ChatGPT-4 (OpenAI, 2023):** We use the GPT-4 model via the ChatGPT API to generate natural language descriptions for synthesized objects.

- **PowerPaint (Zhuang et al., 2024):** PowerPaint V2 is used for context-aware inpainting. It is based on a UNet-like architecture with multi-resolution feature blending and supports both geometric and text-prompted editing. In particular, we adopt the BrushNet (Ju et al., 2024) pipeline from the official PowerPaint implementation, incorporating structure-aware refinement for improved object boundary reconstruction.

- **MoGE (Wang et al., 2024b):** MoGE uses a ViT-L backbone to infer 3D structure from single RGB images. It is pretrained on large-scale image-depth pairs and is used to lift inpainted RGB objects into 3D point cloud space.

- **Grounded-SAM (Ren et al., 2024):** This model integrates Grounding DINO (Swin-T variant) for open-vocabulary object detection and SAM for pixel-accurate segmentation. The combination enables text-driven, high-quality instance mask extraction.

- **InvSR (Yue et al., 2024):** InvSR is a diffusion-inversion-based super-resolution model. It leverages backward sampling from pretrained latent diffusion models to reconstruct high-resolution images from low-resolution observations.

**Models for Multi-Sweep Generation.**    To simulate temporally coherent sequences resembling multi-frame LiDAR scans, we employ additional models:

- **MOFA (Niu et al., 2024):** MOFA animates static object images into temporally coherent video sequences. It uses user-defined 2D tracking points to control motion generation.

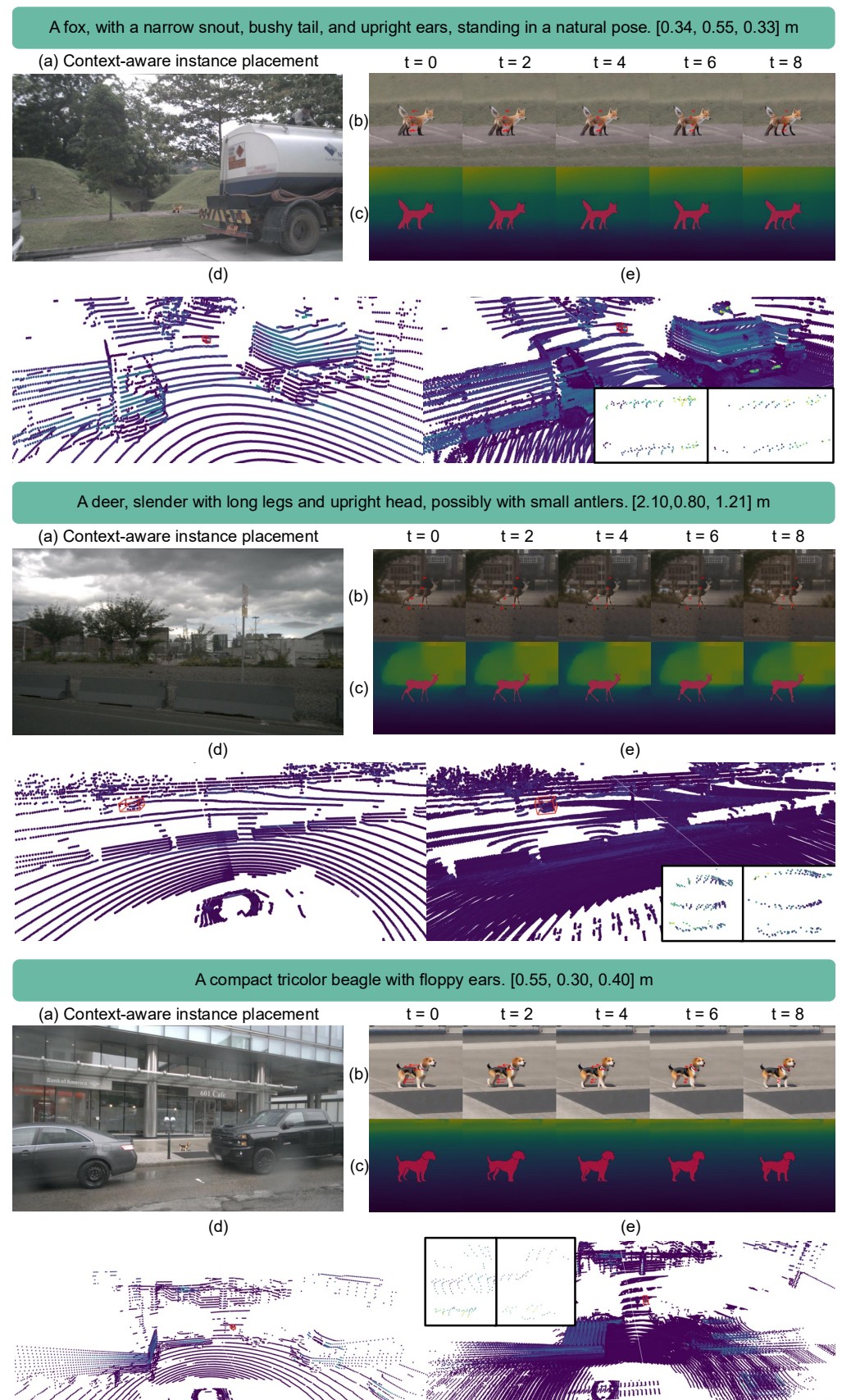

Figure 11: **Examples of animals.**

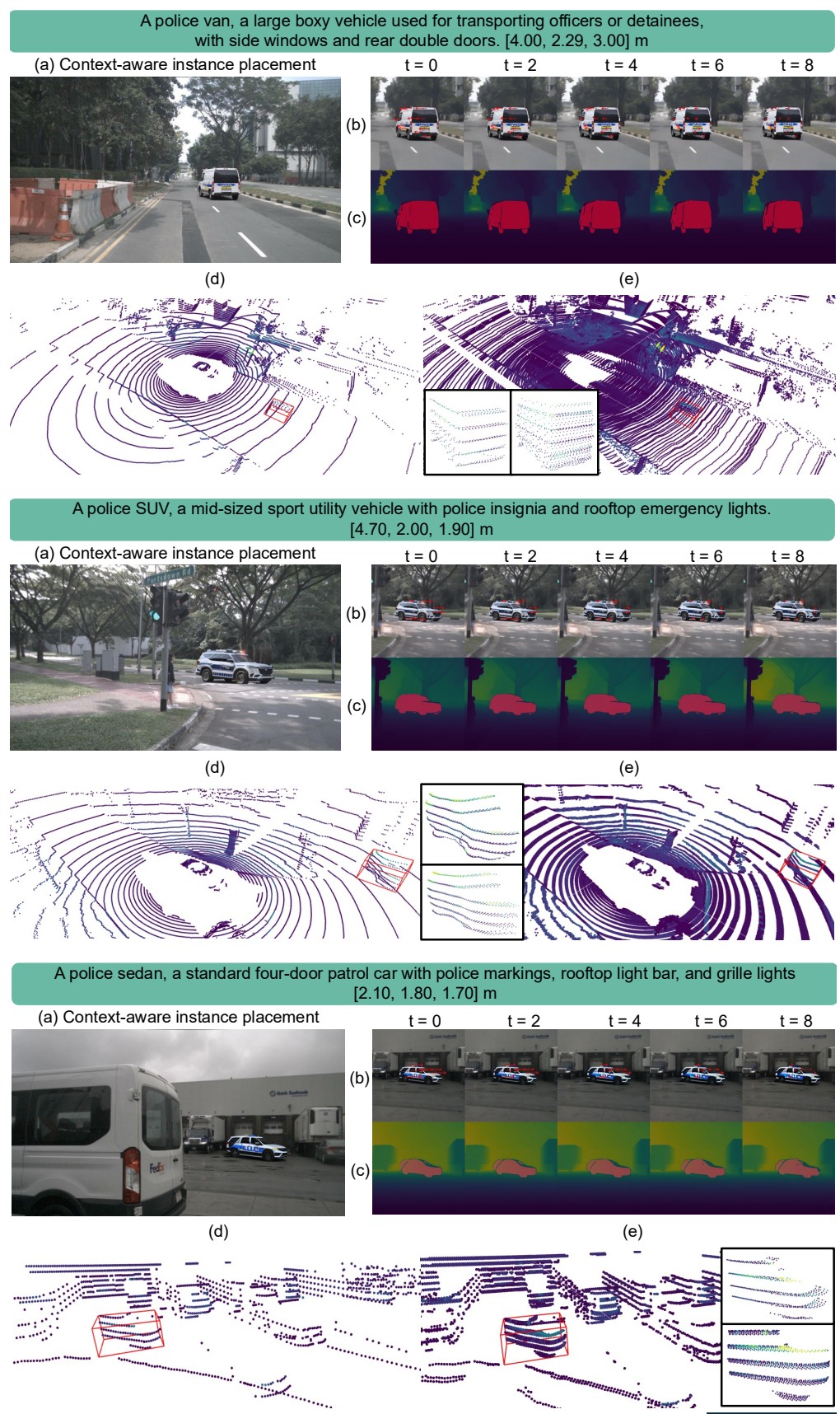

Figure 12: **Examples of police vehicle.**

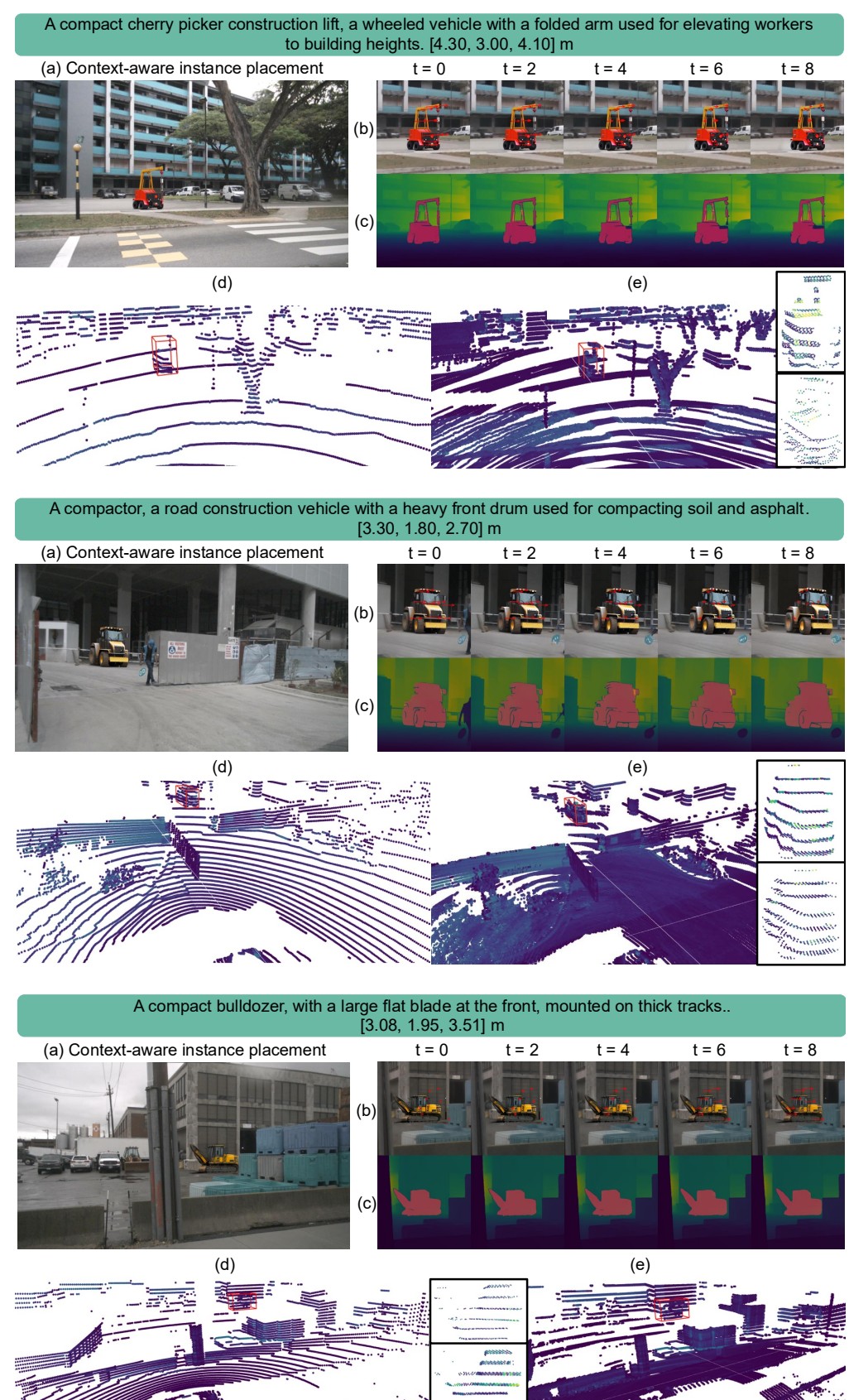

Figure 13: **Examples of construction vehicle.**

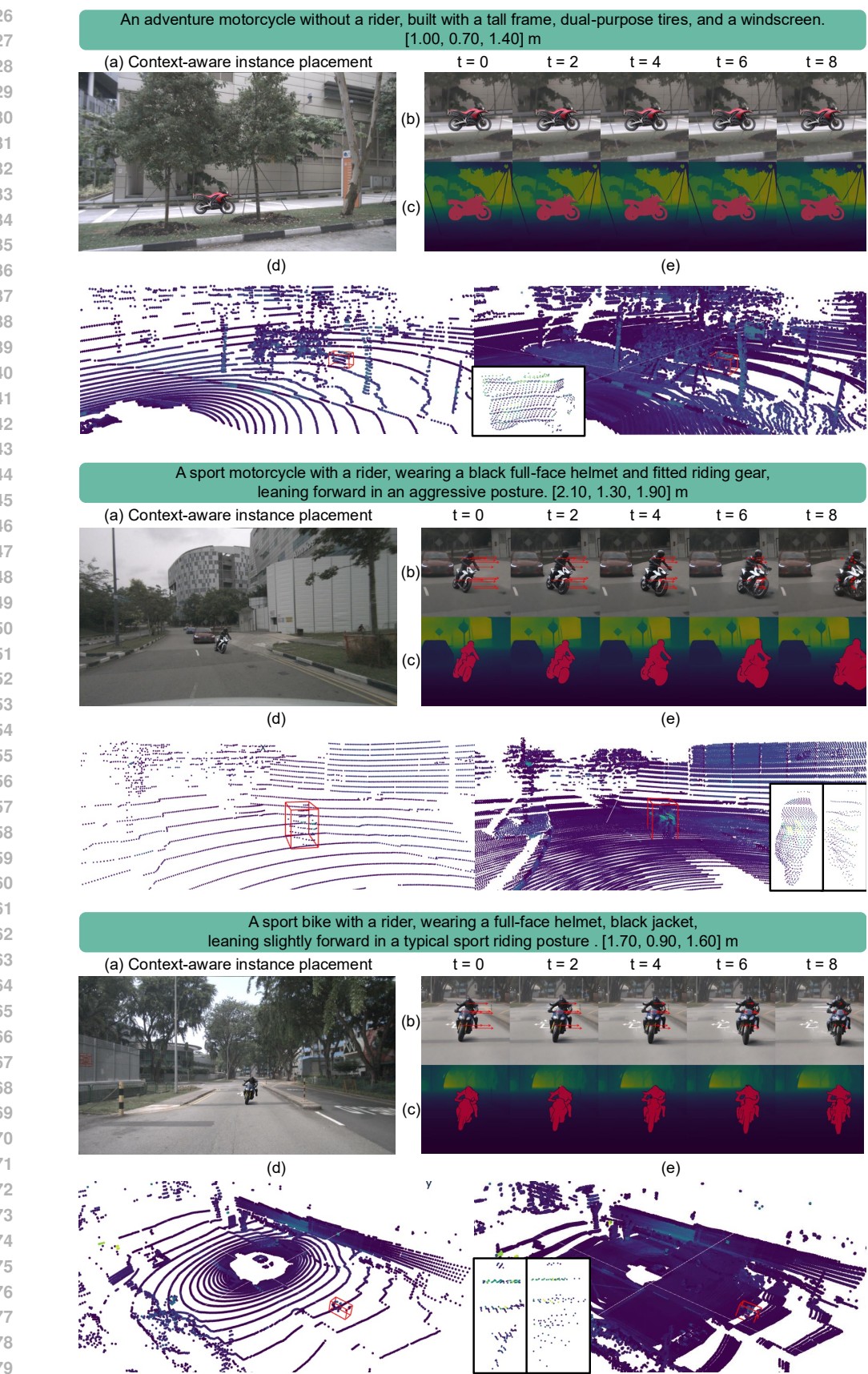

Figure 14: **Examples of motorcycle.**

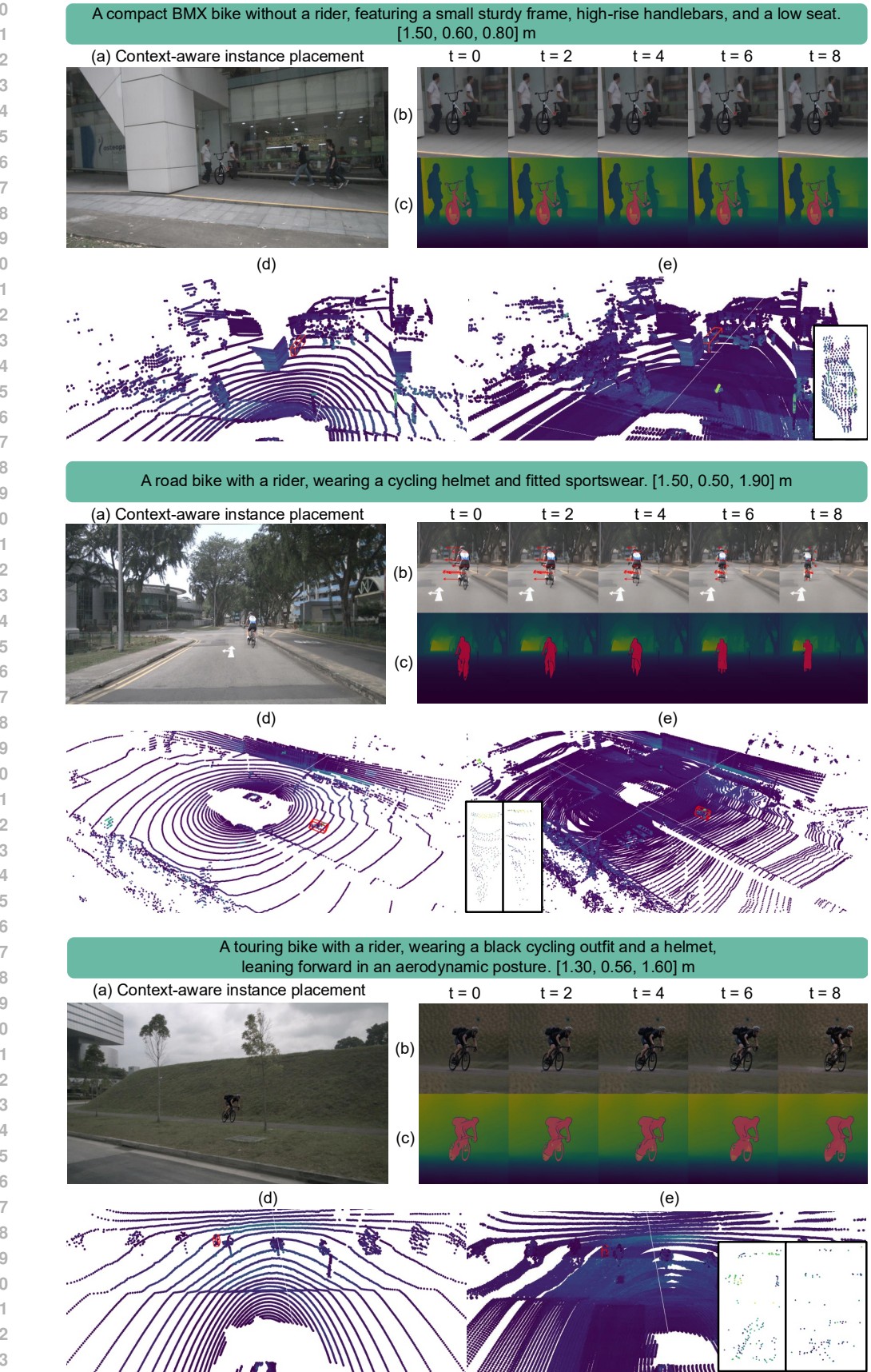

Figure 15: **Examples of bicycle.**

- **Video Depth Anything (Chen et al., 2025):** This model extends Depth Anything V2 to videos, providing temporally consistent monocular depth estimation. We use the ViT-L variant trained on long-sequence consistency objectives. It enables the generation of pseudo-LiDAR sequences from synthetic videos.
- **SAM2 (Ravi et al., 2024) HQ (Ke et al., 2023):** We adopt `sam2.1-hiera-large`, a hierarchical high-resolution segmentation model from Meta AI. We use this model for image and video segmentation to obtain consistent instance masks across frames.

**Computational Cost.** To estimate the cost of semantic verification, we consider the full nuScenes training set comprising 168,780 images. Each cropped region yields approximately 85 visual tokens, concatenated with a prompt of $\sim$30 tokens and an expected output of $\sim$10 tokens. Assuming GPT-4o pricing, the total verification cost is estimated at \$48.52. Although MAPLE integrates multiple foundation models, the entire pipeline remains cost-effective, requiring less than \$50 for semantic verification of the entire nuScenes dataset. This demonstrates the practicality of scaling multimodal augmentation to millions of samples without additional training.

### E.2 RGB Generation with Semantic Verification

**Object Description Generation** To generate realistic subclass-level object descriptions, we employ a large language model (LLM) with carefully designed prompts. Each description contains three key attributes: subclass identity, visual appearance, and approximate physical size $(s_x, s_y, s_z)$. For each target label (e.g., construction vehicle, bicycle, motorcycle), the LLM is instructed to produce one subclass commonly observed in urban environments, together with its visual traits and typical dimensions. Whenever possible, physical sizes are grounded in official manufacturer specifications of a real-world product model. If no exact specification is available, a closely related model is used with the `approximate` flag set to `true`, accompanied by a web search query for traceability. These descriptions not only serve as conditioning for diffusion-based inpainting but also provide priors for subsequent geometric verification.

For bicycles and motorcycles, we explicitly request two sets of bounding box dimensions: one excluding the rider (vehicle only) and one including a seated rider, which significantly changes the aspect ratio and bounding box height. This distinction is crucial for realistic pseudo-LiDAR reconstruction and for training detectors that must handle both rider-free and rider-occupied instances.

The following script shows the exact instruction given to the LLM. The output is required to be valid JSON conforming.

```
LLM Prompt: Subclass Description Generator

You are generating ONE subclass description for data augmentation.

GOAL
- Provide one concrete subclass of {TARGET_LABEL} commonly found in urban
↪  environments.
- Output MUST be valid JSON matching the schema below.
- Use meters for all sizes.

INSTRUCTIONS
1) Choose a realistic subclass (e.g., "bulldozer", "box truck", "commuter
↪  bicycle", "naked motorcycle").
2) Give a concise visual description (colors, shape, notable parts; 1--2
↪  sentences).
3) For size, identify a specific real-world product model from the web
↪  (official/manufacturer spec preferred).
   - Normalize length/width/height to meters.
   - If exact spec is unavailable, pick a close model, set "approximate": true,
   ↪  and include a "web_search_query".
4) If TARGET_LABEL âĹĹ {bicycle, motorcycle}, ALSO provide a second bounding
↪  box that includes a seated rider
   with typical posture. Keep it realistic (do NOT under/over-estimate).

OUTPUT SCHEMA (JSON only, no extra text):
```

```
{
  "target_label": "string",
  "subclass_identity": "string",
  "product_model": "string",
  "visual_description": "string",
  "size_m": { "length": float, "width": float, "height": float },
  "with_rider_bbox_m": { "length": float, "width": float, "height": float } |
  ↪  null,
  "approximate": boolean,
  "source_note": "string",
  "web_search_query": "string"
}

CONSTRAINTS
- JSON must parse. No comments, no trailing commas, no units in numbers.
- Keep "visual_description" factual; avoid brand slogans.
- If bicycle/motorcycle: "with_rider_bbox_m" MUST NOT be null. Else: set null.

Begin now for TARGET_LABEL = {TARGET_LABEL}.
```

Representative outputs are shown in Fig. 3, which illustrate how the LLM generates diverse subclasses within the same category, each grounded by explicit product dimensions.

**Object Inpainting** After obtaining textual descriptions $\mathcal{T}_c$ and spatial masks $\mathcal{M}$ from projected 3D bounding boxes, we perform diffusion-based inpainting to insert novel objects into the RGB image. To preserve scene fidelity, we first extract a local image crop $\mathcal{I}_{\text{crop}}$ centered on the inpainting region $\mathcal{R}_{\text{inpaint}}$. A common challenge is that small or distant objects yield low-resolution crops, which degrade generation quality. To address this, we employ a super-resolution model $f_{\text{SR}}$ (Yue et al., 2024) whenever the shorter side of the crop is below a resolution threshold $R$. This ensures that the subsequent diffusion model operates on high-quality inputs regardless of the native crop size.

Formally, the inpainting process is defined as:

$$\hat{\mathcal{I}} = f_{\text{in}}(\mathcal{I}_{\text{sr}}, \mathcal{T}_c, \mathcal{M}), \quad \mathcal{I}_{\text{sr}} = \begin{cases} f_{\text{SR}}(\mathcal{I}_{\text{crop}}), & \text{if } \text{Res}(\mathcal{I}_{\text{crop}}) < R, \\ \mathcal{I}_{\text{crop}}, & \text{otherwise}, \end{cases} \tag{4}$$

where $\text{Res}(\mathcal{I}_{\text{crop}})$ denotes the shorter side of the crop. The diffusion inpainting model $f_{\text{in}}$ is conditioned on both the description $\mathcal{T}_c$ and the binary mask $\mathcal{M}$, which specifies the exact region to be synthesized. The result is a high-fidelity augmented image $\hat{\mathcal{I}}$ where the inserted object is visually diverse, consistent with the surrounding context, and aligned with the size priors from the description stage. Examples of inpainted results are shown in Fig. 4, demonstrating how MAPLE produces context-aware augmentations with plausible scale, heading, and occlusions.

### E.3 PSEUDO-LIDAR GENERATION WITH GEOMETRIC VERIFICATION

**LiDAR Intensity Simulation** To simulate reflectance, we compute a per-point intensity map $\mathcal{I}_{\text{intensity}}$ based on the grayscale inpainted image $\mathcal{I}_{\text{gray}}$, modulated by surface normal and range-based attenuation (Viswanath et al., 2024; Marcus et al., 2025):

$$\mathcal{I}_{\text{intensity}} = \text{clip}\left(\mathcal{I}_{\text{gray}} \cdot |\mathbf{n}|^{\gamma} \cdot e^{-\alpha|\mathbf{p}|}, 0, 255\right), \tag{5}$$

where $\mathbf{n}$ is the surface normal aligned with the sensor axis, $\gamma$ controls orientation influence, $|\mathbf{p}|$ is the range to point $\mathbf{p}$, and $\alpha$ is the attenuation coefficient. As shown in Fig. 5, the final pseudo-LiDAR is obtained by back-projecting the scaled depth map and semantic mask into 3D space, incorporating both geometry and synthesized intensity.

**Motion Simulation Module for Virtual Object Sweeps** From the initial object point cloud $\mathcal{P}$, we uniformly sample $K$ anchor points using voxel-wise binning along the $x$, $y$, and $z$ axes. These anchor points are temporally propagated according to the object's simulated motion, which models plausible 3D trajectories informed by class-specific velocities. Here, object-specific velocity statistics is sampled from the training set.

1. For bicycle and motorcycle, if the description does not include a rider, we set $(v_x, v_y) = 0$.

2. For animals and police, we query an LLM for typical urban speeds.

For each sweep, we compute the displacement vector $\delta_t$ and apply it uniformly to all sampled anchor points:

$$\delta_t = \mathbf{c}_t - \mathbf{c}_0, \qquad \mathcal{P}_t = \mathcal{P} + \delta_t, \tag{6}$$

resulting in a set of temporally displaced point clouds $\{\mathcal{P}_t\}_{t=0}^{T}$, where $\mathbf{c}_0$ denote the initial center of the object and $\mathbf{c}_t$ be its center at sweep $t$ obtained from motion simulation. These propagated points define coarse motion trajectories that are projected onto the image plane via the projection matrice $\pi$ :

$$\mathbf{u}_t^{(k)} = \pi(\mathbf{x}_t^{(k)}), \qquad \text{for } k = 1, \ldots, K, \tag{7}$$

producing smooth 2D trajectories $\{\mathbf{u}_t^{(k)}\}$ that condition the generation of temporally coherent video frames via the image animation diffusion model.

**Ego-Motion Compensation and LiDAR Formatting** To simulate temporally consistent LiDAR sequences, we apply ego-motion compensation to each sweep and unify them into a common coordinate frame.

Let $\mathbf{T}_t$ denote the ego-pose at time $t$, and $\mathbf{T}_t^{-1}$ its inverse. For each sweep $\mathcal{P}_t$, we perform the following steps:

1. **Ego-frame alignment.** Each point cloud $\mathcal{P}_t$ is transformed from the camera coordinate frame into the ego-centric frame using homogeneous coordinates:

$$\mathcal{P}_t^{\text{ref}} = \left( \mathbf{T}_t^{-1} \cdot [\mathbf{x}_t, 1]^T \right)^T, \tag{8}$$

where a homogeneous coordinate 1 is appended to each point $\mathbf{x}_t$ to support affine transformation. This yields $\mathcal{P}_t^{\text{ref}}$, a point cloud expressed in the unified ego-frame.

2. **LiDAR formatting.** The points are projected onto a simulated 32-channel LiDAR range-view, then unprojected back to 3D space.

3. **Sweep-specific re-transformation.** To simulate the original acquisition conditions, the formatted points are re-transformed back to the original sweep frame using $\mathbf{T}_t$.

The re-aligned point clouds from all sweeps are concatenated to form a temporally coherent pseudo-LiDAR sequence. Additionally, we estimate planar velocity by comparing the mean position across the first and last sweeps for dynamic instances:

$$\mathbf{v} = \frac{\mathbb{E}[\mathcal{P}_T^{(x,y)}] - \mathbb{E}[\mathcal{P}_0^{(x,y)}]}{\Delta t}. \tag{9}$$

This process ensures that our pseudo-LiDAR reflects both temporal motion and ego-vehicle dynamics, providing realistic training signals for motion-aware 3D perception models.

**Bounding Box Refinement.** For each accepted instance, we refine the initial estimate $\mathbf{b}_{3D}$ into a tight bounding box around the synthesized point cloud $\hat{\mathcal{P}}_{\text{lidar}}$, written as $\hat{\mathbf{b}}_{3D} = [\hat{c}_x, \hat{c}_y, \hat{c}_z, \hat{s}_x, \hat{s}_y, \hat{s}_z, \hat{\theta}]$. Here, the center $(\hat{c}_x, \hat{c}_y, \hat{c}_z)$ is taken from the centroid of $\hat{\mathcal{P}}_{\text{lidar}}$, the heading $\hat{\theta}$ is estimated from the dominant eigenvector of the $XY$-plane covariance, and the dimensions $(\hat{s}_x, \hat{s}_y, \hat{s}_z)$ are derived from axis-aligned bounds in the rotated frame and from the vertical extent.

# F  TRAINING SETTING

## F.1  AUGMENTATION STRATEGIES

We consider three instance-level augmentation strategies: GT-Aug (Yan et al., 2018), Text3DAug (Reichardt et al., 2024), and PGT-Aug (Chang et al., 2024). GT-Aug inserts objects sampled from a ground-truth database into training scenes. Text3DAug and PGT-Aug generate synthetic samples by rendering 3D meshes at 13 discrete heading angles (from $-180°$ to $180°$ in $30°$ increments) and

distributing them across a 2D spatial grid covering the full perception range ($[-50\,\text{m}, 50\,\text{m}]$ in both $x$ and $y$, sampled every $5\,\text{m}$). The resulting synthetic instances are stored in a custom database and inserted into training scenes in a 1:1 ratio with ground-truth instances; that is, for each rare class, half of the inserted instances are drawn from the ground-truth database, and the other half from generated samples.

## F.2  3D OBJECT DETECTION

**Instance Augmentation for LiDAR-only Setting.**    For the seven frequent object categories (car, truck, bus, trailer, barrier, pedestrian, traffic cone), we apply GT-Aug (Yan et al., 2018). We insert 7, 6, and 6 instances per scene for the three long-tail categories—construction vehicle, motorcycle, and bicycle. The synthetic augmentation strategies and sampling configurations follow the protocols described in Section F.1. We prioritize inserting our synthesized instances via instance composition to construct each augmented scene. If the number of available synthetic instances for a given category falls short of the per-scene target, we supplement the remainder with ground-truth objects, maintaining consistency with the GT-Aug.

**Implementation Details.**    Following the nuScenes 3D detection benchmark (Caesar et al., 2020), we conduct experiments using the 10-sweep setting. We evaluate our method in the multimodal setting using BevFusion (Liang et al., 2022), which integrates LiDAR and multi-view camera features via a Swin Transformer backbone and a voxel-based encoder, followed by ConvFuser and a TransFusionHead. We follow the original training configuration (Liang et al., 2022), using an AdamW optimizer with a cosine annealing schedule, a base learning rate of 0.0001, and train the model for 6 epochs with a batch size of 3 per GPU. To assess the impact of pseudo-LiDAR augmentation in LiDAR-only scenarios, we additionally report results using CP-Voxel (Yin et al., 2021) and PointPillar Lang et al. (2019). CP-Voxel is a CenterPoint-based detector configured with a fine voxel size of $[0.075, 0.075, 0.2]$, while PointPillar adopts a lightweight PillarVFE encoder with a coarser voxel size of $[0.2, 0.2, 8.0]$. Both models are trained using a one-cycle learning rate schedule, with an initial learning rate of 0.001 and a weight decay of 0.01. The total number of epochs is 20, with a batch size of 8 per GPU. Following common practice in OpenPCDet, we apply a curriculum-style augmentation strategy by disabling instance sampling during the final five epochs to mitigate overreliance on synthetic data and improve generalization to real-world distributions.

## F.3  3D SEMANTIC SEGMENTATION

**Instance Augmentation.**    For the 14 frequent classes, we do not apply any instance-level augmentation. To ensure a controlled comparison, we insert the same number of instances per class in GT-Aug and Text3DAug, PGT-Aug variants as in our method for each training scene, following the sampling and insertion strategy described in Section F.1.

**Implementation Details.**    We follow the nuScenes 3D semantic segmentation benchmark (Caesar et al., 2020) and conduct experiments using the 1-sweep setting. We evaluate our method for 3D semantic segmentation using two architectures: MinkowskiNet (Choy et al., 2019) and SPVCNN (Tang et al., 2020). Both models operate directly on voxelized point clouds with a voxel resolution of $0.05\,\text{m}$. The segmentation head is trained to predict 17 semantic classes following the nuScenes benchmark. MinkowskiNet is a sparse convolutional network based on the MinkowskiEngine framework. SPVCNN follows a similar design but integrates sparse point-voxel convolution layers to enhance efficiency while maintaining high-resolution feature propagation. Both models are trained for 80 epochs using the SGD optimizer with a learning rate of 0.24, momentum of 0.9, and a weight decay of $1 \times 10^{-4}$. We use cosine annealing learning rate scheduling and enable standard augmentation techniques during training, including random rotation, flipping, scaling, transformation, and point dropout. The batch size is 32, with eight workers per GPU for training and validation.

