# OpenReview forum: "MAPLE: Context-aware Multimodal Augmentation for Long-tail 3D Object Detection"
_ICLR.cc/2026/Conference — ICLR 2026 Conference Withdrawn Submission_

### Official Review · Reviewer_Y3zJ · 2025-10-26

**Soundness:** 3
**Presentation:** 3
**Contribution:** 3
**Rating:** 4
**Confidence:** 4

**Summary:**

The paper presents a training-free, context-aware multimodal augmentation pipeline for long-tail 3D detection that inserts rare objects via diffusion-based image inpainting guided by VLM descriptions, reconstructs paired pseudo-LiDAR with depth estimation, and enforces semantic geometric verification alongside a new success-rate reliability evaluation.

**Strengths:**

* The paper frames the long-tail data challenge in 3D perception and the limits of current instance-level, LiDAR-only augmentation.
* The combination of VLM-generated subclass prompts and diffusion inpainting yields objects that blend with scene geometry and occlusions addressing a weakness of 2D/3D placement.

**Weaknesses:**

* Object subclass and physical size are sourced from a VLM, so misclassification or hallucinated dimensions can misguide both inpainting and scaling.
* All downstream results are on nuScenes, external datasets or real-vehicle studies are absent, limiting claims about generalization.

**Questions:**

Additional experiments on external datasets would help validate the effectiveness of the method.

---

### Official Review · Reviewer_kKzu · 2025-10-29

**Soundness:** 3
**Presentation:** 2
**Contribution:** 3
**Rating:** 4
**Confidence:** 4

**Summary:**

The paper introduces MAPLE, a training-free, context-aware, multi-modal instance-augmentation pipeline for long-tail 3D perception. Objects are first inpainted in RGB images using VLM-guided textual subclass prompts. Then, pseudo-LiDAR instances are reconstructed from monocular depth and rescaled to match size priors. Finally, semantic (prompt/region consistency) and geometric (spatial filtering + size-prior) verification remove implausible generations. The authors additionally propose a success-rate evaluation to quantify how verification reduces the end-to-end failure rate. On nuScenes, MAPLE improves both multimodal (BEVFusion) detection and 3D semantic segmentation, and also yields competitive LiDAR-only gains.

**Strengths:**

1. Training-free, modular pipeline composing strong 2D foundation models; no retraining needed.
2. Verification matters: equations + empirical reductions from P(Fnaive) to P(Fverified) (e.g., bicycles 39.7%→2.4%).
3. Zero-shot rare-class recovery (bicycle 8.021 AP with no real bicycles).
4.Cost disclosure and module runtime (useful for practitioners).

**Weaknesses:**

1. Limited scope of evaluation. Only nuScenes is used; no results on Waymo or KITTI. Gains on detection are modest (e.g., +1.10 mAP for BEVFusion; CenterPoint roughly on par), which raises concerns about generality/impact for detection-centric settings.
2. Depth dependence & priors. Pseudo-LiDAR relies on monocular depth plus single-axis size scaling (fixing sz, leaving sx, sy flexible). This can bias aspect ratios and headings; ablations on different depth estimators or λ-range in size-prior checks are absent in the main paper.
3. “First” claim should be tempered. The paper positions MAPLE as the first training-free multimodal augmentation with verification. While the combination is compelling, prior works already explore training-free LiDAR-only augmentation and VLM/diffusion-assisted pipelines; please clarify boundaries of novelty and cite broader contemporaries (e.g., data synthesis pipelines for multi-sensor fusion).
4. The strongest numbers are in segmentation; for detection, motorcycle FID is worse than PGT-Aug (5.10 vs 3.70). A deeper analysis of failure modes (thin structures, spokes, handle bars) would help interpret when MAPLE is preferable.
5. Typos and artifacts. e.g.,:
1) Section title: “ZERO-SHOPT” (should be zero-shot).
2) Page 5: “nstead of” (missing “I”).
3) Section E.2 prompt: “TARGET_LABEL â´L´L {bicycle, motorcycle}” (encoding/∈ issue), etc.

**Questions:**

1. Can the authors report results on Waymo (at least a subset) or KITTI-360 to demonstrate cross-dataset robustness?
2. Please quantify the temporal realism of multi-sweep simulation (e.g., per-sweep Chamfer to real sweeps, object-track smoothness, or optical-flow/scene-flow consistency).
3. How sensitive are results to the depth estimator choice and the λmin/λmax bounds in the size-prior check? Provide ablations.
4. Could the authors release verification prompts and a deterministic seed/config bundle to ensure reproducibility given fast-evolving foundation models?
5. For the zero-shot experiment, what is the precision/recall curve and the number of true positives? The table shows AP=8.021—please add qualitative error analysis.

---

### Official Review · Reviewer_DGLn · 2025-10-29

**Soundness:** 2
**Presentation:** 3
**Contribution:** 2
**Rating:** 2
**Confidence:** 4

**Summary:**

This paper presents a pipeline for augmenting camera-lidar data for autonomous driving scenes, called MAPLE. The pipeline combines a myriad of off-the-shelf foundation models for: VLMs for description generation, image inpainting, mono-to-depth prediction, image-to-video generation, SAM for image segmentation mask, etc.
To insert a new object into a target scene, first a VLM is prompted to suggest a type of object with certain properties, which is then inpainted into the target's scene image. Depth prediction of the image is used to suggests a 3d structure, which can be converted into virtual lidar scans. Image-to-video generation is used for short term object dynamics (e.g. moving limbs), since lidar-based object detection typically requires combining multiple lidar scans.
An final verification step on the geometry and semantics helps identify if the generated result is realistically embedded in the original scene.
In experiments on nuScenes, MAPLE is used to augment infrequent classes: construction vehicles, motor cycles, bicycles. Results show that MAPLE augmentation improves segmentation performance over other data augmentation techniques, and detection results are on par with prior work. Analysis further shows how the spatial distribution of MAPLE's inserted objects is distinct from the grid-like patterns of prior augmentation works.

**Strengths:**

* Data augmentation is a key task to produce robust perception for automated driving; Addressing multi-modal augmentation for synchronized multi-sensor setups is a relevant task.
* The new MAPLE data augmentation technique does not require training, as it relies on powerful foundation models pre-trained for specific tasks which are readily available nowadays. It seems to be a sensible modern approach to data augmentation.
* MAPLE generates both camera and lidar data, including segmentation masks, and also considers real-world details such as combining multiple lidar sweeps, and the motion between those sweeps.
* Based on presented qualitative results, generated samples appear varied, and reasonably realistic. MAPLE includes a geometric verification step to check for realistic results; Analysis of the real and generated data distribution show good overlap with real data (similar to Text3DAug baseline).
* Data augmentation for less common classes with MAPLE brings strong performance improvements on segmentation tasks, compared to baseline augmentation techniques.

**Weaknesses:**

* This pipeline addresses a data-augementation challenge which is relevant for researchers in the autonomous driving application domain. However, the method itself does not provide particularly novel insights on (applied) machine learning itself. I therefore consider this work better suited for a conference related to computer vision or autonomous driving, rather than ICLR.

* If I understand correctly, experiments are performed using OpenAI's GPT-4 using a remote API, and not using open source model parameters. Although the abstract indicates code will be released to support reproducibility, using a commercial closed-source model will not guarantee others could reproduce the results in the future (even with the promised code release). It would have been better to also include experiments using open source VLMs to ensure future reproducibility.

* The paper claims to address long-tail 3D object detection (paper title), but the the prompt in line 174 specifically states "{TARGET_LABEL} commonly found in urban environments", and placement of generated objects is context-aware. As the examples in Figure 4 show, the samples appear to improve intra-class variation and class-imbalance in the training data of typical objects in typical settings, but not really rare objects and/or in rare situations that constitute the really difficult cases (cars on sidewalks, near crashes, furniture on the road, etc.). This doesn't mean the method isn't useful, but claiming it addresses the long-tail seems to be an overclaim.

* Contribution ③, which is about demonstrating the MAPLE method on nuScenes, is not a separate contribution from Contribution ①, the MAPLE method itself. Performing experiments to demonstrate a new method is an expected requirement for a methodological contribution, and thus not a separate contribution by itself.

* The method inserts generated pointclouds into the lidar pointcloud, but the paper does not describe if this process also produces "shadows" behind the generated object, which would hide regions in the scene that have become occluded. If ommitted, it would severely limit the realism of the lidar simulation.

* Experiments, section 5.1: Implementation details are sparse, and more details could be given about how the augmented dataset compares to the original unaugmented one: Overall size, resulting class distribution (vs original), but also practical details as how long it took to generate the augmented data (inc used hardware), etc. As it stands, it is difficult to judge how much effort/time/resources it requires to use MAPLE in practice, and what it took to produce the presented experimental results. It would also have been useful to explore the relation of the amount of MAPLE data augmentation vs performance: at what point does performance plateau, or could we have kept pushing the performance by just adding more of your augmentations?

* Related, for a data augmentation paper, I believe it is important to show that the strategy works on multiple datasets, not only on nuScenes, as then there is a risk that design choices have been optimized to that particular setting.

* Experimental evaluation shows overall mAP/mIoU over all classes, and detailed results for the augmented classes. Unsurprisingly, the performance on these selected classes improves slightly by data augmentation, but it is not clear how performance of the other more common classes (e.g. vehicles) is impacted by the augmentation, if at all.

* Experiments show that in terms of object detection, MAPLE performs on-par (sometimes better, sometimes worse) with prior works, Text3DAug and PGT-Aug. Line 417 explains how these mehods insert objects without scene context, but it isn't clear what other differences the methods have, and thus what can be concluded from the observed (lack of) performance difference.

Minor:
* Figures are referenced out of order, e.g. the first reference is Figure 3 on line 043.
* Line 269, "consistent sequence. nstead" -> typo
* Line 369, "FID" is used here for the first time without any explanation for those unfamiliar with what it means or measures.
* line 188: "(sx, sy, sz) follow priors from Tc.", unclear, how are LLM queries turned into prior distributions?

**Questions:**

* line 207: "Because inpainting attempts are not always successful, we generate multiple candidates and retain only those that satisfy both conditions." -> Unclear: for a single inpainting task, retain all that satisfy (so multiple variants of same scene?), or retain one (wich one)?
* For your experiments, how does the augmented dataset compare to the original one, and how much resources did it require? How does performance scale with the amount of augmented data?
* Beyond the placement distribution, how does MAPLE conceptually compare to Text3DAug and PGT-Aug? Why do the methods perform similarly for object detection, even though MAPLE peforms much better on segmentation?

---

### Official Review · Reviewer_hVZZ · 2025-11-01

**Soundness:** 2
**Presentation:** 2
**Contribution:** 2
**Rating:** 4
**Confidence:** 4

**Summary:**

The paper proposes a training-free pipeline for multimodal instance augmentation to address long-tail 3D object detection. It generates synchronized RGB–LiDAR pairs by inserting objects via context-aware image inpainting and reconstructing pseudo-LiDAR through depth estimation.

**Strengths:**

-- MAPLE inserts objects via diffusion-based inpainting conditioned on VLM-generated descriptions, enabling diverse intra-class variation  not possible with copy-paste or mesh rendering.

-- Semantic verification uses VLM-based QA to filter mislabeled or incomplete inpaintings, reducing naive failure rates

--MAPLE improves mAP for construction vehicles in multimodal detection and mIoU in LiDAR-only segmentation. FID scores confirm pseudo-LiDAR fidelity, reflecting better structural similarity to real data.

**Weaknesses:**

-- The “1,000 human-annotated images” for αsem estimation lack annotation protocol details (e.g., the distribution of annotators).

-- The paper claims “less than $50 for semantic verification” (p. 23) but does not validate this with actual API billing, which is important for scaling data.

-- The method does not compare with some multi-modal data augmentation methods, e.g., [1][2], make some comparisons to such VLM-free based methods is helpful.

-- The overall pipeline seems complicated, make it diffcult for real-world applications.

[1] Exploring data augmentation for multi-modality 3d object detection. 2020

[2] 3d data augmentation for driving scenes on camera. 2024

**Questions:**

Please see the weakness.

---

### Author Response · Authors · 2025-11-27

We sincerely thank the reviewers for their valuable feedback, which has strengthened our work's clarity, reproducibility, and future directions. We especially appreciate the recognition of our three key contributions: (1) multimodal augmentation, (2) semantic/geometric verification, and (3) success-rate evaluation framework.
Rather than addressing individual concerns point-by-point, we use this opportunity to clarify our research motivation, design rationale, and planned improvements. We acknowledge the limitations regarding experimental scope, dataset diversity, and reproducibility, and recognize these as areas for future investigation.

**Motivation from Long-tail Problem in LiDAR Dataset**

Our research addresses the question: "Can we automatically synthesize rare objects that become increasingly sparse as LiDAR datasets adopt fine-grained ontologies?" While 3D perception systems encounter diverse objects in deployment, existing datasets are biased toward frequent classes, limiting systematic long-tail evaluation. As reviewers noted, this explains why our experiments focused on nuScenes and did not cover extreme scenarios (e.g., vehicles on sidewalks, debris on roads, objects in unusual poses before collision). Given these constraints, we augmented existing long-tail distributions with synthetically generated rare-class samples.

KITTI and Waymo define only three coarse categories, precluding long-tail observation. NuScenes defines 23 classes and retains meaningful imbalance even after consolidation to 10 categories, making it the primary benchmark for object-level long-tail distribution. Existing LiDAR-centric methods (GT-Aug, Text3DAug, PGT-Aug) use copy-paste or mesh synthesis but remain limited by asset libraries.

We designed an image-first pipeline for two reasons. First, diffusion-based RGB synthesis generates diverse appearances, poses, and contextual placements from text alone, substantially expanding intra-class variation. Second, given that modern systems rely on RGB–LiDAR fusion, generating RGB first and reconstructing paired LiDAR via depth estimation provides unified multimodal augmentation. While this introduces depth errors and alignment challenges, verification mechanisms mitigate these issues. Future work will extend validation to additional benchmarks and, as reviewers suggested, report performance on non-augmented classes.

**Evaluation Challenge and Verification-Centric Approach**

An equally critical challenge is the inherent difficulty in reliably evaluating performance on rare categories. In most autonomous driving datasets, rare classes lack sufficient samples to produce stable evaluation metrics. For instance, among nuScenes' 23 fine-grained object classes, ambulance and animal appear fewer than several dozen times in the entire validation split, failing to provide distributional coverage across diverse subclasses, poses, scales, distances, and headings. This issue becomes more severe at the point-level. Due to LiDAR's distance-dependent characteristics, point density decreases sharply for distant or thin objects, preventing rare classes from acquiring sufficient geometry to reliably define 3D bounding box dimensions and orientations even when they do appear. Consequently, rare-class evaluation suffers from dual constraints—insufficient sample counts and incomplete geometry—rendering GT-based metrics (AP, mIoU) inadequate for reliably measuring augmentation effectiveness.

These limitations in GT-based evaluation provide important design implications for augmentation frameworks. When direct comparison to GT is unstable, alternative quality criteria become necessary: (1) cross-modal consistency among generated multimodal data, (2) alignment with learned priors about real-world objects, and (3) intermediate metrics that directly assess pipeline reliability. MAPLE adopts three verification strategies based on this understanding. First, cross-modal consistency verification enforces geometric alignment across the RGB generation → depth estimation → pseudo-LiDAR reconstruction pipeline. Second, prior-based validation encompasses both semantic and geometric verification. Semantic verification employs VLMs to assess whether generated objects satisfy semantic correctness and visual plausibility relative to real-world instances, while geometric verification uses category-specific size priors and spatial filtering to confirm physical plausibility of reconstructed pseudo-LiDAR and remove depth artifacts. Third, our success-rate evaluation framework quantifies not only downstream task performance but also the proportion of valid samples produced at each pipeline stage, enabling indirect assessment of augmentation quality and pipeline stability even under GT scarcity. In essence, MAPLE is a verification-centric framework that evaluates quality based on "how reliably the generation process maintains multimodal consistency and real-world priors" rather than "similarity to GT."

---

> ### Author Response · Authors · 2025-11-27
>
> **Ablation Study**
>
> - Depth Dependency
> : While MAPLE does rely on monocular depth estimation, we do not view this as a fundamental limitation because the bounding box corresponding to the inpainting region is already provided, making local structural preservation more critical than absolute depth accuracy. Empirically, prior work [1] has demonstrated performance gains even with noisy pseudo-LiDAR, and Table 7 confirms that geometric verification adequately compensates for depth errors. Nevertheless, following the reviewers' suggestion, we will include ablation experiments with alternative depth estimators.
>
> - Scaling Behavior
> : To address the request for clearer analysis of performance trends with varying data quantities, we will add scaling experiments with augmentation ratios of 0.5×, 1×, 2×, and 4×. This will demonstrate the point at which MAPLE performance plateaus.
>
> - Open-source LLM
> : We acknowledge the reproducibility concerns arising from versioning and updates in closed LLMs, as reviewers noted. To address this, we will conduct ablation studies using open-source VLMs under identical pipeline configurations, and release all prompts and verification scripts to strengthen reproducibility.
>
> - VLM Error Propagation
> : The concern that errors in LLM-generated subclass descriptions or size estimates may propagate downstream is valid. MAPLE mitigates this through two-stage filtering via semantic and geometric verification. As shown in Table 2, failure rates decrease substantially—from 52.6% to 10.4% for Construction Vehicles and from 39.7% to 2.4% for Bicycles—indicating that most VLM errors are filtered out proactively. However, overly strict prior filtering risks removing rare but valid variations; we will include additional experiments analyzing this trade-off.
>
> - $\alpha_{sem}$ Estimation Details
> : $\alpha_{sem}$ was estimated based on a single annotator performing three repeated evaluations over one month on 1,000 samples, a design intended to ensure intra-annotator consistency. We acknowledge potential inter-annotator variance as reviewers noted, and will augment our manuscript with brief analysis and protocol clarification.
>
> - Related Work
> : The papers [2, 3] mentioned by reviewer hVZZ employ copy-and-paste strategies using existing GT instances rather than synthesizing new data via generative models as MAPLE does. We will incorporate these works into our related work section and clarify the methodological distinction.
>
> We hope this work can contribute to establishing design principles and evaluation methodologies for training-free composable augmentation pipelines. Once again, we deeply appreciate the reviewers' invaluable feedback.
>
> [1] Chang et al., "Just add $100 more: Augmenting pseudo-lidar point cloud for resolving class-imbalance problem", NeurIPS 2024
>
> [2] Exploring data augmentation for multi-modality 3d object detection. 2020
>
> [3] 3d data augmentation for driving scenes on camera. 2024

---

### Note · Authors · 2025-12-03

I have read and agree with the venue's withdrawal policy on behalf of myself and my co-authors.